# The Role of Sex in the Effect of Vocal Attractiveness on Ultimatum Game Decisions

**DOI:** 10.3390/bs13050433

**Published:** 2023-05-20

**Authors:** Junchen Shang, Chang Hong Liu

**Affiliations:** 1Department of Medical Humanities, School of Humanities, Southeast University, Nanjing 211189, China; 2College of Psychology, Liaoning Normal University, Dalian 116029, China; 3Department of Psychology, Bournemouth University, Poole BH12 5BB, UK

**Keywords:** vocal attractiveness, fairness of offer, sex difference, ultimatum game

## Abstract

The present research investigated the role of sex in the effect of vocal attractiveness on fairness judgment in a two-person Ultimatum Game. Each participant in the game decided whether to accept offers from a proposer who was either associated with an attractive or unattractive voice. The results showed that while participants were more likely to accept fair offers, they would also accept some unfair offers that were associated with an attractive voice. This effect of vocal attractiveness was more clearly shown by female participants, although all male and female participants took longer to make a decision when an attractive voice was associated with an offer, regardless of whether the voice was from the same sex or the opposite sex. Overall, the results inform the role of sex in the effect of vocal attractiveness and further confirm the beauty premium effect on economic bargaining, where people with an attractive voice would benefit.

## 1. Introduction

The preference for physical attractiveness has been a phenomenon of widespread interest to both psychologists and economists. From an evolutionary perspective, physical attractiveness plays an important role in mate selection, because of its putative mate value about good health [1,2]. Physical attractiveness also has strong influences on human behavior in various social environments outside of mating contexts, such as the labor market and economic decision-making (see [3] for a review). The preference for individuals with attractive faces in economic activities is known as the “beauty premium”, which has been extensively examined in behavioral research [4,5,6] and neuroscience [7,8,9]. For example, participants are more likely to accept unfair offers from a proposer who has an attractive face in a bargaining game [8,10]. They also tend to take longer to refuse an unfair offer from a person with an attractive face [8], although they refuse an unfair offer more quickly if it comes from a person with an unattractive face [9]. These findings suggest that attractive faces have reward value and can reduce people’s negative feelings of unfairness [8,9]. 

Attractiveness is not limited to a person’s face but can also be shown in a person’s voice and scent [11]. Compared to numerous studies on the role of facial attractiveness in decision-making, relatively little research has examined the effects of vocal attractiveness on economic decision making. A handful of recent studies in this vein have shown that attractive voices can also create beauty premium effects similar to facial attractiveness [12,13,14], where unfair offers were more easily accepted when they were proposed by an attractive voice [12], and trustees with an attractive voice were also more likely to receive investment [13,14]. 

However, research to date has not investigated whether there are sex differences in the effect of vocal attractiveness. Since vocal attractiveness is correlated with mate selection [11], the attractiveness of opposite-sex voices should have a stronger influence on behavior than that of same-sex voices. Casual observations also appear to support this. For example, in China, a lot of men prefer to use the voice of the female star Lin Chi-ling for navigation software rather than the voices of male stars (https://m.163.com/dy/article/EH2QREMD054521ZY.html [7 June 2019], https://www.weekinchina.com/2014/02/route-to-success/ [21 February 2014]). On the other hand, Jacky Cheung, a male singer, has charmed countless Chinese women with his voice (https://www.163.com/dy/article/GQQ2ET8G0541UIB2.html [9 December 2021]). Hence, it is likely that the effect of vocal attractiveness on fairness judgments is different between same-sex voices and opposite-sex voices. Shang et al. [12] only examined the effects of male voices and did not compare the responses of male and female participants in their analysis. Although Shang and Liu [13,14] compared the effects of male and female voices, they found no differences between them. However, there is evidence that the sex of the participants can interact with the sex of the stimuli. For example, whether the participant is of the same sex or the opposite sex as the target face can modulate the effect of facial attractiveness in decision-making [3]. Therefore, it is important to include both variables in this line of research.

Some studies also included sex in the analysis of the effect of facial attractiveness in experimental economic games. However, the findings have not been consistent. For example, although Solnick and Schweitzer [6] observed the main effects of attractiveness and participant sex in an ultimatum game, neither face sex nor participant sex interacted with facial attractiveness. In contrast, Rosenblat [15] observed that women (but not men) offered more money to attractive recipients in a dictator game when both the faces and voices of recipients were presented, although the sex of recipients did not matter. However, when only the faces or voices of recipients were presented, no effect of facial attractiveness was found. It is worth noting that in Rosenblat’s study [15], only facial attractiveness (but not voice attractiveness) was rated. A study by Lucas and Koff [16] also found a beauty premium effect displayed by female participants at high or low risk for conception. However, unlike Rosenblat [15], they found the effect was stronger for male than female recipients in both ultimatum and dictator games. Moreover, when acting as responders, female participants at low conception risk (but not those at high conception risk) accepted more unfair offers from attractive allocators of both the same sex and the opposite sex. The use of fertility status in their design may be the reason why Lucas and Koff’s [16] results differed from some other studies [6,15].

However, Li and Zhou [17] also reported different results about the role of sex without using fertility status as a factor. They found effects for both participant and proposer sex. Their participants acted as bystanders who were not involved in money splits in a dictator game. Results showed that the participants punished attractive proposers of the same sex more severely than attractive proposers of the opposite sex. They also punished unattractive proposers of the opposite sex more severely than unattractive proposers of the same sex. However, when the participants were asked to rate the reasonableness of the proposers’ offers, neither facial attractiveness nor proposer sex were significant.

Given these results, Voit et al. [18] suggested that differences in the economic game may lead to conflicting results in sex differences in the beauty premium. They tested whether and how the sex of the participant and the sex of the partner moderated the beauty premium in four different games. In the dictator game and the prisoner’s dilemma game, both male and female participants offered more money and cooperated more when attractive partners were of the opposite sex than when they were of the same sex. However, in the ultimatum game, male participants showed a greater beauty premium effect than female participants for both female and male partners. In the trust game, regardless of participant sex and partner sex, attractive partners received more money than unattractive partners. With regards to the discrepant findings in the literature, the authors noted that they recruited participants aged 18 to 58 years old and used young to middle-aged partner faces, while most other studies [6,15,16,17] only recruited young university students. Hence, the differences in age may be a cause of the dissimilar findings. Taken together, however, sex appears to play a role in decision making with partners of variable facial attractiveness, although the effects may vary in different economic decision-making situations.

If sex interacts with facial attractiveness in some economic games [15,16,17], would it also modulate the beauty premium of vocal attractiveness in economic decision-making? Based on the previous finding that evaluations of facial attractiveness ratings are correlated with evaluations of vocal attractiveness [19,20] and the finding that both facial and vocal attractiveness are correlated with mating preferences [11], the answer should be yes. To test this hypothesis, the present study investigated the effect of sex on the beauty premium effect of vocal attractiveness in a two-person ultimatum game adopted from Shang et al. [12]. The ultimatum game uses a simplified bargaining context in the lab, excluding complex factors in the real bargaining environment, so as to disentangle the dynamics of the bargaining decision [6]. We used the game to examine the influence of the vocal attractiveness of proposers on the bargaining process of responders towards various degrees of distribution fairness, where both the effects of target (proposer) sex and participant (responder) sex were measured. If sex plays a role in the beauty premium of vocal attractiveness, male participants should be more affected by female vocal attractiveness, whereas female participants should be more affected by male vocal attractiveness.

## 2. Method and Materials

### 2.1. Participants

We used MorePower 6.0.4 to estimate the required sample size for a 2 × 2 × 2 × 5 mixed factorial design with one between-participant factor and three within-participant factors. To obtain a large effect with η_p_^2^ = 0.14, a sample size of at least 52 participants was required at α = 0.05 with 80% statistical power [21]. Therefore, we recruited 58 students (31 females, *M*_age_ = 22.48 years, *SD* = 2.33 years) from Liaoning Normal University for this study. All participants had normal or corrected-to-normal vision and normal hearing, and all reported being physically and mentally healthy. The research was conducted in accordance with the Declaration of Helsinki and approved by the Institutional Review Board of Liaoning Normal University. Written informed consent was obtained from each participant before the experiment.

### 2.2. Design and Materials

This study used a 2 (sex of voice: male, female) × 2 (vocal attractiveness: attractive, unattractive) × 2 (participant sex: male, female) × 5 (offer fairness: 5:5, 4:6, 3:7, 2:8, 1:9) mixed factorial design, where participant sex was a between-participant factor and sex of voice, vocal attractiveness, and the offer fairness were within-participant factors. Offer fairness was defined by a ratio, which ranged from completely fair (5:5) to extremely unfair (1:9). The dependent variables were proportional acceptance and response times.

Voice stimuli were the same neutral-valence vowels used in Shang and Liu [14], which were originally adopted from Ferdenzi et al. [22]. Each voice consisted of three vowel syllables (/i/, /a/, /ou/). The duration of all voices was adjusted to 2040 ms with Praat software v.5.3.85. The sound intensity was adjusted to 70 dB. We used 30 female voices and 30 male voices in the experimental trials. Among each sex, 15 had attractive voices and 15 had unattractive voices. The attractiveness of the voices was determined based on ratings from 58 participants (on a 7-point scale from 1 = “very unattractive” to 7 = “very attractive”) in Shang and Liu [14]. A 2 (vocal attractiveness: attractive, unattractive) × 2 (sex of voice: male, female) ANOVA was performed on the ratings. This showed a significant main effect of attractiveness, *F*(1, 56) = 362.55, *p* < 0.001, η_p_^2^ = 0.87, while no significant result was found for the main effect of sex, *F*(1, 56) = 0.40, *p* = 0.531, or the interaction between the two factors, *F*(1, 56) = 2.15, *p* = 0.148. Attractive female voices (*Mean* = 5.15, *SD* = 0.31) were judged as more attractive than unattractive female voices (*Mean* = 2.81, *SD* = 0.49). Similarly, attractive male voices (*Mean* = 4.91, *SD* = 0.38) were judged as more attractive than unattractive male voices (*Mean* = 2.91, *SD* = 0.55). For the practice trials, we used two attractive and two unattractive voices that were not used in the experimental trials.

### 2.3. Procedure

Participants were tested individually in a sound-attenuated room. The voices were presented binaurally via Sennheiser HD206 headphones. The experiment was controlled using E-prime version 2 (Psychology Software Tools, Pittsburgh, USA) on a HP 280 Pro G2 MT computer. Instructions and stimuli were presented on a 19-inch LCD monitor with a refresh rate of 60 Hz and a screen resolution of 1440 × 900 pixels.

We adopted the two-person ultimatum game task from previous research on economic decision-making [8,12]. An explanation of the game was given to the participants. Participants were told that the voices and offers were taken from a previous study, and they would bargain with some real proposers. They were also told that the payment for their participation was CNY 30 plus the gain of two randomly selected trials.

Figure 1 illustrates the trial procedure with an example. Each trial began with a fixation presented for 400–600 ms on the center of the screen, followed by a 200–300 ms interval. A voice of the allocator was subsequently presented for 2040 ms, also followed by a 200–300 ms blank interval. A proposal on how to split CNY 10 between the proposer and the participant (responder) was then presented. The example in the figure shows that the proposer proposes to allocate CNY 7 to themself and CNY 3 to the participant. This proposal remained on the screen until the participant responded. The participant had to decide whether to accept or reject the offer by pressing a key. Half the participants were told to press the “A” key to accept and the “L” key to reject, while the remaining half were told to press the “L” key to accept and the “A” key to reject. If an offer was accepted, the participant and the allocator would receive the amount of money as proposed; otherwise, both of them would receive nothing. Once a key press response was made, there was a 200–300 ms interval, followed by the income outcome from this trial presented on the screen for 2000 ms. After this, the instruction “the next trial will begin” was shown for 200–300 ms before continuing to another trial.

Five different offers (CNY 1, CNY 2, CNY 3, CNY 4, and CNY 5) were presented to each participant with 60 voices during the experimental trials. Each voice was paired with one of the offers, and each pair was repeated twice. This amounted to a total of 120 experimental trials, in which each offered condition consisted of 24 trials. Within the 24 trials for each offered 12 trials were presented with a male voice and the remaining 12 trials with a female voice. Furthermore, half of the voice stimuli in each sex were attractive, and the other half were unattractive. The presentation of the 120 trials was random for each participant.

Prior to the experimental trials, participants were given 20 practice trials to become familiar with the task. The allocations in practice trials were not included in the participants’ rewards.

After the experiment, the participants rated the attractiveness of the 60 voices on a 7-point scale (from 1 = “very unattractive” to 7 = “very attractive”).

## 3. Results

The mean proportion acceptance results and the response time results are shown in Table 1. Repeated-measures ANOVAs were performed on the two measures. The Greenhouse–Geisser correction was applied for sphericity departures. All post hoc analyses were Bonferroni-corrected.

### 3.1. Proportion Acceptance

Significant main effects were found for vocal attractiveness, *F*(1, 56) = 15.84, *p* < 0.001, η_p_^2^ = 0.22; offer, *F*(2.63, 147.51) = 196.41, *p* < 0.001, η_p_^2^ = 0.78; and sex of voice, *F*(1, 56) = 8.21, *p* = 0.006, η_p_^2^ = 0.13. These effects were qualified by three significant two-way interactions. Other main effects or interactions were not significant.

To identify the source of the two-way interactions, we conducted simple effect analyses. First, for the interaction between vocal attractiveness and participant sex (see Figure 2A), *F*(1, 56)= 6.76, *p* = 0.012, η_p_^2^ = 0.11, the analysis showed that female participants accepted more offers following attractive voices than unattractive ones, *F*(1, 30) = 17.30, *p* < 0.001, η_p_^2^ = 0.37, whereas male participants accepted a similar number of offers regardless of voice attractiveness, *F*(1, 26) = 1.48, *p* = 0.236, η_p_^2^ = 0.05. Second, for the interaction between sex of voice and sex of participant illustrated in Figure 2B, *F*(1, 56) = 5.88, *p* = 0.019, η_p_^2^ = 0.10, the analysis showed that male participants accepted more offers following a female voice than a male voice, *F*(1, 26) = 11.58, *p* = 0.002, η_p_^2^ = 0.31, whereas female participants accepted a similar number of offers regardless of the sex of the voice, *F*(1, 30) = 0.12, *p* = 0.74, η_p_^2^ = 0.004. Lastly, for the interaction between vocal attractiveness and offer, shown in Figure 2C, *F*(3.13, 175.11) = 2.94, *p* = 0.033, η_p_^2^ = 0.05, the analysis showed that although acceptance levels were largely determined by the fairness of the offer, more unfair offers were accepted when they were associated with an attractive voice rather than an unattractive one. Specifically, this attractiveness effect was observed at the offer levels 3:7, 2:8, and 1:9, *F*s ≥ 5.28, *p*s ≤ 0.025, but not at the fair offer levels (5:5 and 4:6), *F*s ≤ 3.62, *p*s ≥ 0.06.

### 3.2. Response Time

The main effect of vocal attractiveness was significant, *F*(1, 56) = 4.90, *p* = 0.031, η_p_^2^ = 0.08, where decisions were slower for attractive voices (*M* = 1171, *SD* = 462) than for unattractive (*M* = 1104, *SD* = 451) voices. The main effect of offer was also significant, *F*(2.79, 156.44) = 12.15, *p* < 0.001, η_p_^2^ = 0.18, where decisions for the fairest offer (5:5) were faster than for all other levels of fairness, *p*s < 0.01, while the decision speeds for other fairness levels were comparable, *p*s ≥ 0.06. All other main effects or interactions were not significant, *F*s ≤ 2.42, *p*s ≥ 0.06.

### 3.3. Post-Experiment Ratings of Vocal Attractiveness

A 2 (vocal attractiveness: attractive, unattractive) × 2 (sex of voice: male, female) ANOVA on the mean attractiveness ratings of the voices after the main experiment found a significant main effect of attractiveness, *F*(1, 56) = 190.47, *p* < 0.001, η_p_^2^ = 0.77, where attractive voices (*M* = 4.77, *SD* = 0.41) were rated as more attractive than unattractive voices (*M* = 2.79, *SD* = 0.67). No significant results were found for the main effect of sex, *F*(1, 56) = 0.08, *p* = 0.783, or the interaction between sex and attractiveness, *F*(1, 56) = 1.47, *p* = 0.23. This confirms that the pre-experiment selection of voices was effective.

## 4. Discussion

The main purpose of this study was to examine the role of sex in the effect of vocal attractiveness. Our first result revealed a gender difference between the male and female responses. As illustrated in Figure 2A, only female participants showed a clear tendency to accept offers more often associated with an attractive voice. Because this effect did not interact with the sex of the voice, it means that the female participants were equally affected by the attractive male and female voices. The male participants, on the other hand, tended to accept offers following a female voice, although their decisions were not affected by the female’s vocal attractiveness (see Figure 2B).

It is worth noting that although both sex and fairness levels played a role in the effect of vocal attractiveness on fairness judgments, sex did not interact with the degrees of fair distribution. The lack of a three-way interaction suggests that the effect of sex was not influenced by the degree of fair distribution. This may seem puzzling because distribution fairness did modulate the effect of voice attractiveness in our results. The relationship between the three variables is a question that calls for further research.

Some aspects of these results are consistent with prior research. For example, Rosenblat [15] also observed a similar sex difference, where the beauty premium of facial attractiveness was only found in female participants when both the faces and voices of opponents were presented, whereas the effect was not found in male participants. It is worth noting that Rosenblat [15] only measured facial attractiveness but not the vocal attractiveness of opponents and did not find an effect of facial attractiveness when only the faces or voices of opponents were presented. We should be cautious about comparing our findings with Rosenblat’s [15]. Consistent with our male participants’ results, Eckel and Grossman [23] also found that male participants were more likely to accept offers from females. They termed this phenomenon “chivalry”, which referred to men’s willingness to help women. Some studies also showed that both male and female participants were more likely to cooperate with female partners in economic games [6,24,25].

From an evolutionary perspective, facial and vocal attractiveness are correlated with mating preferences [11]. We were also interested in comparing our findings with the research on facial attractiveness, although it is not appropriate to make direct comparisons because the effect of facial attractiveness on decision-making may be different from that of vocal attractiveness. In contrast to prior research [17,18], which found participants responding to facial attractiveness differently depending on whether the face of their opponents was of the same sex or opposite sex in the dictator game and prisoner’s dilemma game, our male participants responded comparably to attractive female and male voices. Our female participants also did not treat sex differently. However, participants in different economic games could use different strategies, which may affect sex differences in the decisional process [26]. In fact, Voit et al. [18] only found an effect of opposite sex in two of their three tasks. They did not find any effect in the ultimatum game. Additionally, the discrepancies between the present study and the studies using faces [17,18] could be due to the different processing of vocal attractiveness and facial attractiveness. Further research is necessary to examine whether our findings in the ultimatum game are transferable to other decision environments. It is also worth noting that we used non-word syllables in this study, but bargaining in real life involves meaningful speech from the opponent. Hence, the lack of evidence for the effect of sex on the beauty premium of vocal attractiveness could be due to this limitation. 

Overall, our results demonstrate that although participants clearly preferred fair offers to unfair offers, their decisions could be swayed by attractive voices, particularly when a moderately unfair offer was accompanied by an attractive voice (see Figure 2C). This was the case regardless of whether the voice was from the same or different sex of the participant. Shang and Liu [13,14] also reported a similar effect. These results reinforce the idea that a preference for an attractive voice could alleviate the feeling of dissatisfaction with unfair money distribution [12]. 

As with acceptance rates, response times were also affected by vocal attractiveness, where decisions were slower following an attractive voice relative to an unattractive voice. Unlike acceptance rate, however, this effect did not interact with participant sex or offer fairness, meaning that both male and female participants were equally influenced by vocal attractiveness regardless of whether the offer was fair or unfair or whether the voice was from a male or female. The effect could be attributed to the greater attention caused by an attractive voice relative to an unattractive voice. Unsurprisingly, our participants were quicker to respond to fair offers than to unfair offers.

## 5. Conclusions

In sum, the current study demonstrates that female participants are more likely to accept offers associated with an attractive voice, whereas male participants are more likely to accept offers associated with female voices regardless of their vocal attractiveness. Additionally, both male and female participants are more likely to accept an unfair offer following an attractive voice, regardless of whether the voice comes from a male or female. Participants would even accept a highly unfair offer (e.g., 3:7, 2:8, 1:9) if it was preceded by an attractive voice rather than an unattractive voice. Furthermore, both male and female participants took a longer time to respond to an offer when the offer was associated with an attractive voice, irrespective of the sex of the voice. These findings suggest a complex role for sex in the effect of vocal attractiveness.

## Figures and Tables

**Figure 1 behavsci-13-00433-f001:**
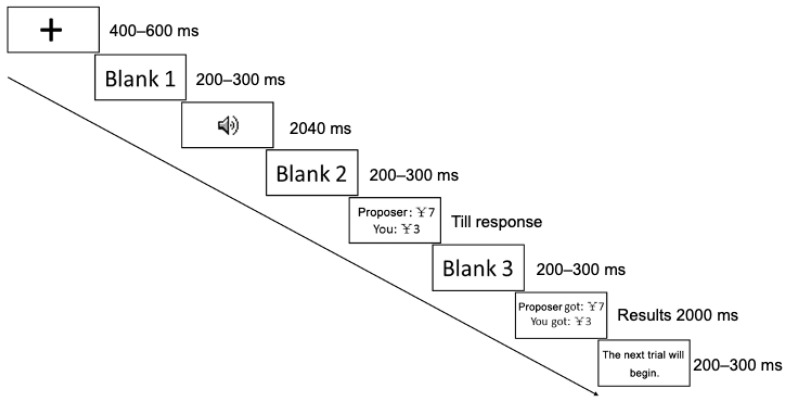
Example of a trial procedure. An attractive or unattractive voice was presented after a fixation and a blank screen. After another blank screen, a proposer’s offer was displayed until the participant responded by pressing a key to accept or reject the offer. After this, the payoff was shown on a blank screen.

**Figure 2 behavsci-13-00433-f002:**
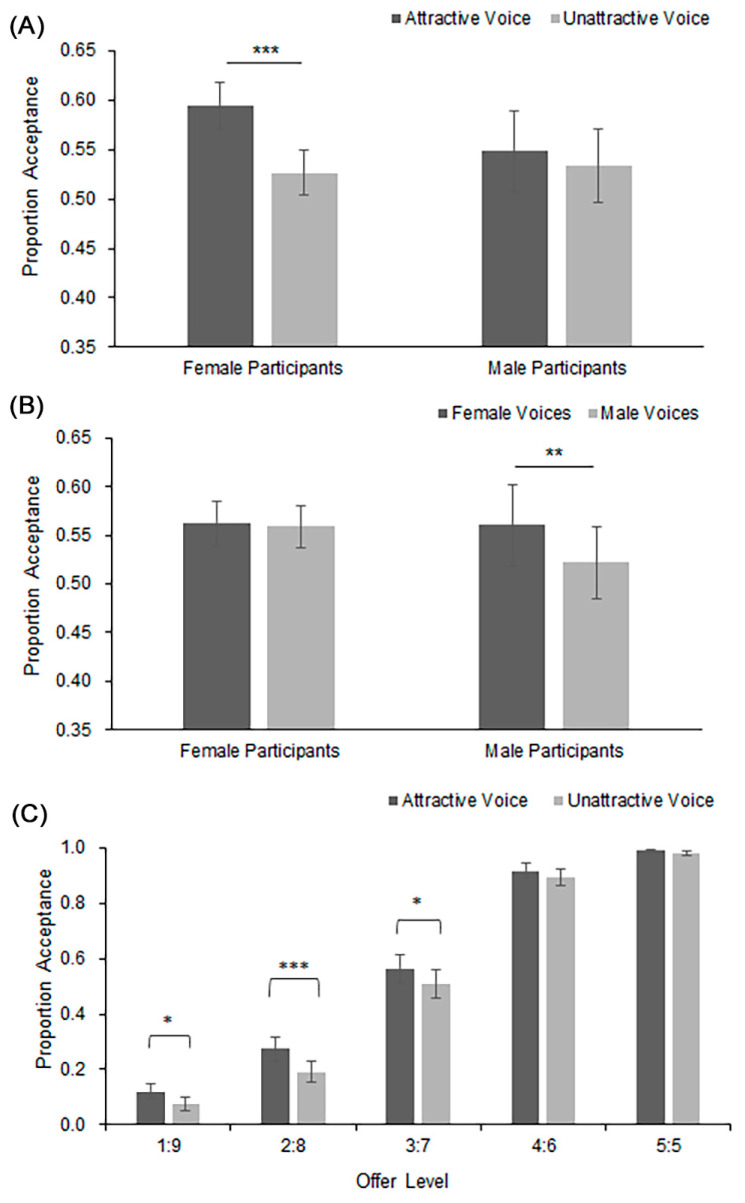
(**A**) Mean proportion of offer acceptance as a function of participants’ sex and vocal attractiveness. (**B**) Mean proportion of offer acceptance as a function of participants’ sex and sex of voice. (**C**) Mean proportion of offer acceptance as a function of offer fairness and vocal attractiveness. Error bars represent one standard error about the mean. * *p* < 0.05, ** *p* < 0.01, *** *p* < 0.001.

**Table 1 behavsci-13-00433-t001:** Mean proportion of acceptance and mean response time (ms) as a function of vocal attractiveness, offer, sex of participant, and sex of voice (standard deviations are shown in parentheses).

Proposer	Participant	Offer after Attractive Voice					Offer after Unattractive Voice				
		5:5	4:6	3:7	2:8	1:9	5:5	4:6	3:7	2:8	1:9
		Mean proportion of acceptance									
Female	Female	0.99	0.96	0.60	0.33	0.06	0.98	0.90	0.54	0.19	0.05
		(0.03)	(0.11)	(0.36)	(0.33)	(0.16)	(0.07)	(0.24)	(0.38)	(0.30)	(0.12)
	Male	0.99	0.90	0.51	0.28	0.18	0.98	0.88	0.52	0.23	0.14
		(0.03)	(0.28)	(0.45)	(0.39)	(0.30)	(0.10)	(0.29)	(0.42)	(0.38)	(0.28)
Male	Female	0.99	0.94	0.65	0.26	0.15	0.98	0.92	0.49	0.17	0.03
		(0.03)	(0.15)	(0.38)	(0.33)	(0.27)	(0.09)	(0.15)	(0.36)	(0.26)	(0.06)
	Male	0.98	0.87	0.47	0.22	0.09	0.99	0.86	0.48	0.18	0.09
		(0.06)	(0.29)	(0.45)	(0.35)	(0.25)	(0.06)	(0.29)	(0.43)	(0.28)	(0.22)
		Mean response time									
Female	Female	820	1144	1429	1303	1091	863	1074	1377	1265	1022
		(374)	(591)	(797)	(688)	(683)	(398)	(654)	(886)	(810)	(498)
	Male	838	1127	1314	1325	1319	845	1025	1242	1431	1139
		(256)	(560)	(698)	(861)	(1089)	(293)	(384)	(647)	(923)	(559)
Male	Female	804	1137	1253	1294	1182	785	1116	1104	1189	1016
		(385)	(690)	(637)	(697)	(670)	(338)	(752)	(588)	(609)	(516)
	Male	1254	1275	1227	1253	1066	853	1025	1310	1090	1347
		(1317)	(1042)	(595)	(713)	(644)	(391)	(402)	(701)	(602)	(1183)

## Data Availability

The data are available from the corresponding authors upon reasonable request.

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
