# Peer review of "The Role of Sex in the Effect of Vocal Attractiveness on Ultimatum Game Decisions"

_behavsci, 2023, doi:10.3390/bs13050433_

Round 1
Reviewer 1 Report
1. In The Method and Materials - Participants: the design was mentioned as factorial 2 x 2 x 2 x 5 (offer of fairness: 5:5, 4:6, 3:7, 2:8, 1;9). With regards to the last factor (offer of fairness, because every participant received all condition and the level of treatment was not a primary interest of the research, this cannot be considered as a factor in the design. Therefore the factorial design is 2 x 2 x 2.
2. Considering that the first two factors (sex of voice and vocal attractiveness) actually were presented as repeated measure, we may question order effects in the results. However, I could not find explanation regarding counterbalancing procedure, if it was performed. If not, it is advisable to discuss the potential biases from order effect in the discussion of results from this experiment.
Author Response
Response 1: Thank you for this comment. We included offer fairness as a within-participant factor in the design because we anticipated an interaction between this factor and voice attractiveness as previously reported by Shang et al. (2021). We acknowledge that we could have explained this more clearly. As Reviewer 3 also suggests that we discuss the interaction between this and other factors, we have now explained the reason for studying this factor in the final paragraph of Introduction: “We used the game to examine the influence of vocal attractiveness of proposers on the bargaining process of responders towards different degrees of fair distributions”. Furthermore, we have also discussed relationship between offer of fairness and other factors in more detail in the Discussion section (see p.8). We hope that these clarifications adequately address the concerns
Response 2: Sorry for the confusion. To clarify, the sex of the voice and vocal attractiveness were both presented as repeated measures in a random order across the 120 trials for each participant (see p.5, line 211, highlighted). This common randomization method was employed to minimize potential order effects. We hope that you also agree that this method was effective.
Reviewer 2 Report
The authors study the "beauty premium" in a bargaining setting, The novelty of their approach lies in the association between vocal attractiveness and sex of partcipants. In this context, the authors build on the existing literature and extend the main results of previous papers in the field.
The experiment is nicely constructed and the basic research question is interesting.
I have only one main problem with this research. The sample size (58 subjects) is really too small. Even if (as the authors claim) the size is in line with some of the previous studies, the protocol on the minimum number of subjects required for experimental investigations is always evolving and I believe authors should comply with this norm.
I also believe that small samples sizes are not idel for hypotheses testing, the results can be affected.
Therefore, my strong advice is that the authors increase considerably the size of the sample, before publication.
Author Response
Response: Thank you for your suggestion. We used MorePower 6.0.4 to estimate the required sample size for our 2×2×2×5 mixed factorial design, which had one between-participant factor and three within-participant factors. To achieve a large effect of ηp2 = .14, we needed a sample size of at least 52 participants at α = .05 with 80% statistical power (Campbell & Thompson, 2012). Our sample size was determined by this power analysis. We have now added this information on the minimum number of subjects required to the participants section. We hope this new justification of our sample size is satisfactory.
Reviewer 3 Report
The role of sex in the effect of vocal attractiveness on fairness judgement is investigated in this research within a two-person Ultimatum Game. The manuscript is well organized and clearly presented.
1. The manuscript title as well as the first sentence in Abstract needs to be revised since it cannot reflect the role of sex in the effect of vocal attractiveness.
2. Real-world cases are strongly recommended to be added in Introduction to show the role of sex in the effect of vocal attractiveness on fairness judgement.
3. The role of sex in the effect of vocal attractiveness on fairness judgement is also related to the degrees of fair distribution, which should be briefly discussed in this research.
Author Response
Response 1: Thank you for your suggestion. We have now changed manuscript title been to “The Role of Sex in the Effect of Vocal Attractiveness on Ultimatum Game Decisions”.
We have also changed the first sentence in Abstract to “The present research investigated the role of sex in the effect of vocal attractiveness on fairness judgement in a two-person Ultimatum Game”.
Response 2: Thank you for the suggestion. We have followed your suggestion to add real-world cases to show the role of sex in the effect of vocal attractiveness on behavior in the introduction (see paragraph 3): “Since vocal attractiveness is correlated with mate selection (Groyecka et al., 2017), the attractiveness of opposite-sex voices should have a stronger influence on behavior than same-sex voices. Casual observations also appear to support this. For example, in China, a lot of men prefer to use the voice of the female star Lin Chi-ling for Navigation software rather than the voices of male stars. On the other hand, Jacky Cheung, a male singer, has charmed countless Chinese women with his voice. Hence, it is likely that the effect of vocal attractiveness on fairness judgements is different between same-sex voices and opposite-sex voices.
Response 3: Thank you for the suggestion. We have added a brief discussion on p.8 about the relationship between degrees of fair distribution and the role of sex in the effect of vocal attractiveness on fairness judgement: “It is worth noting that although both sex and fairness level played role in the effect of vocal attractiveness on fairness judgements, sex did not interact with the degrees of fair distribution. The lack of a three-way interaction suggests that the effect of sex was not influenced by the degrees of fair distribution. This may seem puzzling because distribution fairness did modulate the effect of voice attractiveness in our results. The relationship between the three variables is a question that calls for further research.”
Round 2
Reviewer 2 Report
The authors provide a clever answer to my point: they do not increase the sample size but they prove that the size does not negatively affect the significance of the results.
Still, the sample is extremely small and it cannot be poved whether the results (which, overall, are partly confused) could be clearer, using a larger sample.
However, the authors have improved significantly the presentation, and now the paper is more professionally written, the research's methodology is in line with the main literature in the field. Furthermore, the authors do find an interesting difference between man and women, with man responding more to attractive female voices, whilst women are less sex-oriented, even though, also for them, voice's attractiveness is still very important in bargaining games.